# Surgical Success Rate of Scleral Buckling Surgery and Postoperative Incidence of Cystoid Macular Edema: 10 Years of Experience at a Single Academic Hospital

**DOI:** 10.3390/jcm11185321

**Published:** 2022-09-10

**Authors:** Amaka Watanabe, Masahiro Ishida, Asuka Takeyama, Yoshikazu Ichikawa, Akio Mizushima, Yutaka Imamura

**Affiliations:** 1Department of Ophthalmology, Teikyo University School of Medicine, University Hospital Mizonokuchi, 5-1-1 Futako Takatsuku, Kawasaki 213-8507, Japan; 2Department of Palliative Medicine, Juntendo University Graduate School of Medicine, Bunkyo-ku, Tokyo 113-8421, Japan; 3Sakurashinmachi Eye Clinic, Setagaya-ku, Tokyo 154-0015, Japan; 4Department of Ophthalmology, Toho University Ohashi Medical Center, Meguro-ku, Tokyo 153-8515, Japan

**Keywords:** cystoid macular edema, rhegmatogenous retinal detachment, scleral buckling surgery

## Abstract

The purposes of this study were to report the surgical success rate of scleral buckling surgery (SB) for rhegmatogenous retinal detachment (RRD) in a single academic hospital, analyze the incidence of cystoid macular edema (CME) after SB using optical coherence tomography (OCT), and reveal the factors associated with the development of CME. This was a retrospective observational study of patients with RRD who underwent SB from 2010 to 2020 in a single academic hospital. The anatomical success was initially achieved in 267 of 288 eyes (92.7%), and finally achieved in 288 eyes (100%). After excluding 17 eyes that underwent vitrectomy for reoperations, a total of 271 eyes of 267 patients (173 men; age, 43.5 ± 16.9 years) were retrospectively analyzed to evaluate the incidence of postoperative CME. CME occurred in 6 of 271 eyes (2.2%) within 3 months after initial surgery. Pseudophakic and aphakic eyes appeared more likely to develop CME (chi-squared test: *p* = 0.0078). Five of the six cases with postoperative CME were able to be medically treated. Scleral buckling surgery showed a high success rate even in the era of small-gauge vitreous surgery, and the postoperative frequency of CME after SB was low (2.2%). Previous cataract surgery may be associated with the development of postoperative CME, which is mostly medically manageable.

## 1. Introduction

Scleral buckling surgery (SB) is a procedure for rhegmatogenous retinal detachment (RRD) that seals retinal breaks and relieves vitreous traction to retinal tears using scleral buckles. Although small-gauge vitrectomy with high-speed cutters recently tends to be chosen as a first choice for RRD, scleral buckling is still an effective modality, particularly for RRD in young patients and in cases with small and peripheral breaks [1,2]. This procedure has a high success rate; however, there are infrequent reports of complications [3]. Cystoid macular edema (CME) can occur after scleral buckling surgery and has mostly been studied using fluorescein angiography (FA), in which the accumulation of dye in a cystoid pattern is observed. CME can now be diagnosed using optical coherence tomography (OCT), and FA is not always required [4]. CME can cause visual disturbance and metamorphopsia; therefore, it is important to know the clinical characteristics of CME after buckling surgery in the era of OCT. Here, we report the surgical outcomes of SB for RRD over 10 years and the incidence of postoperative CME in a single academic hospital.

## 2. Materials and Methods

### 2.1. Ethical Approval

This retrospective study was approved by the Institutional Review Board of Teikyo University School of Medicine and conformed to the tenets of the Declaration of Helsinki. Informed consent for surgery and measurements was obtained from each of the patients.

### 2.2. Patient Selection

This was a retrospective and observational study of patients with RRD who underwent scleral buckling surgery at Teikyo University School of Medicine, University Hospital in Mizonokuchi (Kanagawa, Japan) from June 2010 to October 2020. All patients underwent comprehensive ophthalmic preoperative examinations, including the measurement of best-corrected visual acuity (BCVA), intraocular pressure, slit-lamp examination, fundus examination, and OCT. Spectralis OCT (Heidelberg, Germany) was used to evaluate the macula. Previous histories related to CME, including diabetic retinopathy, uveitis, and retinal vein occlusion, were excluded.

### 2.3. Surgical Procedures

Surgery was performed by 4 retinal surgeons (M. Ishida, A. Takeyama, Y. Imamura, or Y. Ichikawa). Retinal breaks were identified in all patients and were treated with transscleral cryotherapy. Scleral buckling surgery was performed using silicon exoplants. Cryotherapy was used in all the patients to close retinal breaks, and diathermy was not used in our institution. Intravitreal gas injection, mostly with air, was generally used in cases of upper retinal breaks, particularly when subretinal fluid remained between breaks and a buckle. Subretinal fluid drainage was performed in cases with much fluid where buckles could not effectively seal retinal breaks. Encircling was chosen in cases with multiple breaks, in which segmental buckles could not seal all. All surgeries were performed under retrobulbar anesthesia, and the patients mostly received the same postoperative medications for 3 months, including topical antibiotics and prednisolone acetate. Patients underwent examinations similar to the preoperative examinations at 1 week, 1 month, 3 months, and 1 year after scleral buckling surgery. BCVA was converted to the logarithm of the minimal angle of resolution (logMAR). CME was defined by the presence of cystic spaces in the outer nuclear layer, outer plexiform layer, and/or in the inner nuclear layer on OCT (Figure 1 and Figure 2), which was observed until 1 year after surgery.

### 2.4. Statistical Analysis

Numerical data were summarized with means and standard deviations, and categorical data were summarized with percentages and frequencies. The chi-squared test was used to compare the frequency of CME. *p*-values of <0.05 were considered to indicate statistical significance. All statistical analyses were performed using JMP^®^ 11 (SAS Institute Inc., Cary, NC, USA).

## 3. Results

Among 298 eyes in 294 consecutive patients who underwent scleral buckling surgery, 10 eyes were excluded because preoperative OCT images had not been taken. Anatomical success was initially achieved in 267 of 288 eyes (92.7%) and finally achieved in 288 eyes (100%). After excluding 17 eyes that underwent vitrectomy for reoperations, a total of 271 eyes of 267 patients (176 eyes in 173 men and 95 eyes in 94 women) were retrospectively analyzed to evaluate the incidence of postoperative CME (Table 1). The age of the patients ranged from 8 to 91 years (mean age, 43.5 ± 16.9 years), and the mean preoperative logMAR BCVA was 0.28 ± 0.55. Two hundred forty-seven eyes (91.1%) were phakic; one eye (0.4%) was aphakic; and twenty-three eyes (8.5%) were pseudophakic. Among the pseudophakic eyes, 19 eyes had intracapsular intraocular lens (IOL), 3 with extracapsular IOL and 1 with sutured IOL. A total of 2 patients had undergone cataract surgery within 4 months before buckling surgery, while 21 had undergone cataract surgery more than 4 months before buckling surgery.

Retinal detachment was macula-off in 109 eyes (40.2%) and macula-on in 162 eyes (60.0%). A single break was found in 146 eyes (53.9%); two breaks were found in 63 eyes (23.2%); three breaks were found in 32 eyes (11.8%); and more than four breaks were found in 30 eyes (11.1%). We used either segmental buckling or encircling; a total of 259 eyes (95.6%) received segmental buckling, and 12 eyes (4.4%) received encircling buckling (#506, 255 eyes; #509, 10 eyes; #286, 3 eyes; #287, 8 eyes; #240, 9 eyes; #270, 9 eyes; #511, 3 eyes; MIRA Inc., Waltham, MA, USA). Subretinal fluid (SRF) drainage was performed for 166 eyes (61.3%); a total of 229 eyes (84.5%) were treated with electrocautery and 42 eyes (15.5%) with 27-gauge needles. Intraocular tamponade with air was performed in 52 eyes (19.2%).

Representative images of a case that underwent successful SB are shown in Figure 1. The final anatomical success rate was 100% either for cases only with scleral buckling surgery or cases both with scleral buckling surgery and vitrectomy. The postoperative logMAR BCVA was 0.08 ± 0.33 and was significantly improved in comparison with the preoperative value (*p* < 0.001).

CME occurred in six (2.2%) eyes in six patients, none of whom had diabetes. Regarding the lens status, 3 of 247 phakic eyes (1.2%) and 3 of 24 (12.5%) pseudophakic and aphakic eyes developed CME (chi-squared, *p* = 0.0078) (Table 2). The detailed clinical characteristics are shown in Table 3, while images of two representative cases are shown in Figure 2. There were no significant differences in other factors, including the macular status, SRF drainage, air tamponade, or the performance of encircling and laser in situ keratomileusis (LASIK) (all *p* > 0.05). Furthermore, there were no significant differences in the number of breaks, extent of retinal breaks, or extent of retinal detachment (all *p* > 0.05). Eyes with a history of cataract surgery appeared more likely to develop CME. Since the number of the cases with CME was low, we did not perform a multivariate regression analysis to identify predictors for CME.

**Representative Cases****.** Case 1 was a 23-year-old woman who underwent scleral buckling surgery for RRD with a superior temporal atrophic hole associated with lattice degeneration. The fovea in her left eye was slightly detached. Subretinal drainage was not performed. The original hole was sealed on the buckle, and the subretinal fluid was gradually absorbed. CME appeared 1 month after buckling surgery. CME temporarily disappeared with oral acetazolamide. However, when internal use was withdrawn, CME recurred. Recurrent CME was observed three times in 7 months. CME disappeared completely after three recurrences. The logMAR BCVA improved from 1.22 to 0.

Case 2 was a 48-year-old-man who had undergone cataract surgery for his left eye 5 years before the appearance of retinal detachment. He underwent scleral buckling surgery for macula-off RRD due to a nasal tear in his left eye. We performed air tamponade without draining the subretinal fluid. The subretinal fluid was gradually absorbed; however, CME was observed 2 weeks after buckling surgery. The CME did not improve with acetazolamide treatment; thus, he was treated with the subtenon injection of triamcinolone acetonide and the intravitreal injection of anti-vascular endothelial growth factor. CME slightly persisted; however, the logMAR BCVA improved from 0.52 to 0.22.

The treatment courses of CME for the other cases are summarized below. Case 3 presented with CME 1.5 months after SB. After 1 month of oral kallidinogenase and topical prednisolone acetate, CME disappeared. Case 4 had CME 1 month after SB. After 2 weeks of oral acetazolamide and topical NSAIDs, CME did not disappear, and after 4 months of the treatment with topical prednisolone acetate, it disappeared. Case 5 had CME 3 months after SB. After 4 months of the treatment with topical prednisolone acetate, CME disappeared. Case 6 showed CME 3 months after SB. After 1 month of the treatment with topical prednisolone acetate, it disappeared.

## 4. Discussion

We reported the surgical results of SB, investigated the incidence of CME after SB using OCT, and found that eyes with a history of cataract surgery were associated with the development of CME. Initial anatomical success was achieved in 267 of 288 eyes (92.7%) and finally achieved in 288 eyes (100%). The final anatomical success rate was 100% either for cases only with scleral buckling surgery or cases both with scleral buckling surgery and vitrectomy. A recent study comparing the surgical outcomes of SB and pars plana vitrectomy for simple phakic macula-on retinal detachment reported that the initial success rate of SB for RRD was 94.9% (130/137) and that of vitrectomy was 84.7% (116/137) [5]. Their study included patients aged <65 years and diagnosed with uncomplicated phakic macula-on primary RRD, which was different from our study. However, considering that our initial success rate was 92.7%, which included patients aged 65 years or older and those with pseudophakic and macula-off cases, it seems that surgical efficacy of SB is not inferior to that of vitrectomy. Mete et al. showed that CME after vitrectomy occurred in 6/29 patients (20.7%) in the macula-off group and in 5/30 (16.7%) in the macula-on group, with no difference in CME incidence between the groups (*p* = 0.69) [6]. We found CME in 6 of 271 patients (2.2%), a rate that seems to be lower than their report.

We did not encounter severe complications in our patients except for redetachment; however, CME was infrequently observed. To our knowledge, the report by Lai et al. is the only study to have examined the incidence of CME using OCT [7]. Their report demonstrated significant differences between patients with and without CME, noting that CME was associated with older age, more extended RD, macular detachment, and external drainage [7]. In our study, there were no significant differences in those factors; however, the incidence of CME was significantly greater in pseudophakic and aphakic eyes than those without.

Sabates et al. reported that macular complications were discovered in 27% of patients who had undergone scleral buckling surgery, with the most frequent complication being CME (16%) [8]. Tunc et al. indicated that scleral buckling surgery was a risk factor for postoperative CME and that it was associated with higher rates of CME than pneumatic retinopexy [9]. Furthermore, CME developed more frequently in patients with a detached macula before surgery (*p* = 0.03) [9]. In this study, CME occurred in 6 of 271 eyes (2.2%); thus, the incidence was lower than those reported in previous studies. Previous studies reported that the incidence of CME was 5–29% in phakic eyes, 27–50% in pseudophakic eyes, and 40–60% in aphakic eyes [8,9,10,11]. Our results showed that the incidence of CME was 1.2% in phakic eyes, 13.0% (3/23) in pseudophakic eyes, and 0% (0/1) in aphakic eyes in patients who had undergone successful scleral buckling. Less invasive SB surgery appears to be important to lower CME incidence.

Meredith et al. reported that among eyes that had been aphakic for more than 4 months before new scleral buckling surgery for RRD, 13 of 24 eyes (54.0%) developed CME [10]. Lobes et al. found no associations between the rate of CME development after scleral buckling surgery and the duration for which the eyes had been pseudophakic or aphakic [12]. Furthermore, preoperative macular detachment, the duration of detachment, the surgical technique, and refraction were not significantly associated with the incidence of CME [12]. In this study, three pseudophakic eyes which developed CME had received cataract surgery from 7 months to 5 years before buckling surgery. The duration from cataract surgery to buckling surgery appeared not to be associated with the development of CME.

The exact cause of CME after successful detachment surgery has not been elucidated; however, Miyake stated that in the postoperative eye, the capillaries were apparently more susceptible to the breakdown of the blood–retinal barrier and showed increased permeability due to prostaglandins [3]. A recent study investigating iris vasculature changes following SB in eyes with RRD with anterior-segment optical coherence tomography angiography revealed a uniform reduction in the iris vessel network for the entire 6-month follow-up after SB [13]. Intravitreal changes in the cytokines released due to ischemic changes following SB may induce CME. Typically, topical nonsteroidal anti-inflammatory drugs (NSAIDs), subtenon injections or intravitreal injections of triamcinolone acetonide, and intravitreal injections of anti-vascular endothelial growth factor (anti-VEGF) are used for the treatment of CME. A previous study reported that postoperative CME completely resolved after an intravitreal injection of a dexamethasone implant [14], and Lai et al. reported that the remission of CME was observed in five of nine patients after subtenon injections of triamcinolone acetonide [7]. We treated CME with topical NSAIDs, topical prednisolone acetate, subtenon injections of triamcinolone acetonide, intravitreal injections of bevacizumab, or oral acetazolamide and kallidinogenase; however, CME persisted in one of six eyes. CME occurred from 2 weeks to 3 months after scleral buckling surgery and resolved within 16 months in five of the six eyes. Regardless of a history of cataract surgery, we prescribed topical antibiotics and prednisolone acetate in this study. According to our findings, the patients who had undergone cataract surgery tended to exhibit more inflammation than those who had not undergone cataract surgery. Therefore, as suggested in the recent investigations by Shorstein et al. [15], it might be better to use topical NSAIDs and prednisolone acetate after scleral buckling surgery for the prevention of CME.

The present study had some limitations. We are aware that different surgical interventions to heterogenous pathologies of RRD may have affected the occurrence of CME. We used OCT and not FA to identify CME. The number of cases of CME in this study was relatively small in comparison with previous studies, and the occurrence of CME in pseudophakic and aphakic eyes might have been coincidental, which makes the statistical power in this study limited. We could not perform multivariate regression analysis to elucidate predictors for CME because the number of CME cases was too small. However, the surgical success rate and incidence of CME in a single institution over 10 years appeared to give us information of the benefits and disadvantages of SB very accurately.

## 5. Conclusions

In summary, we found that scleral buckling is still an effective modality for RRD in the era of small-gauge vitrectomy with less complications and that CME occurs infrequently in patients undergoing scleral buckling surgery. Previous cataract surgery appears to be a risk factor for the development of CME, which is mostly manageable with topical medication.

## Figures and Tables

**Figure 1 jcm-11-05321-f001:**
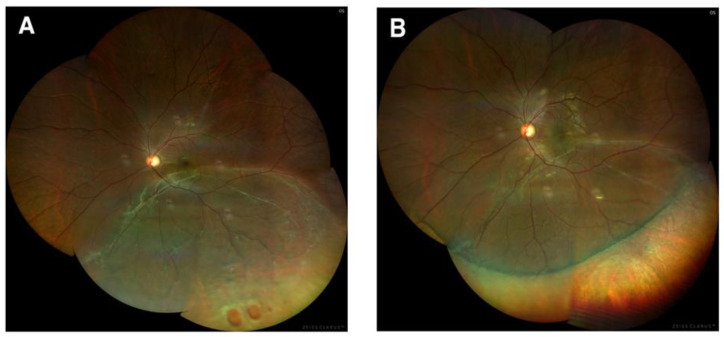
A 32-year-old woman showed rhegmatogenous retinal detachment with a peripheral break (**A**) and was successfully treated with scleral buckling surgery (**B**).

**Figure 2 jcm-11-05321-f002:**
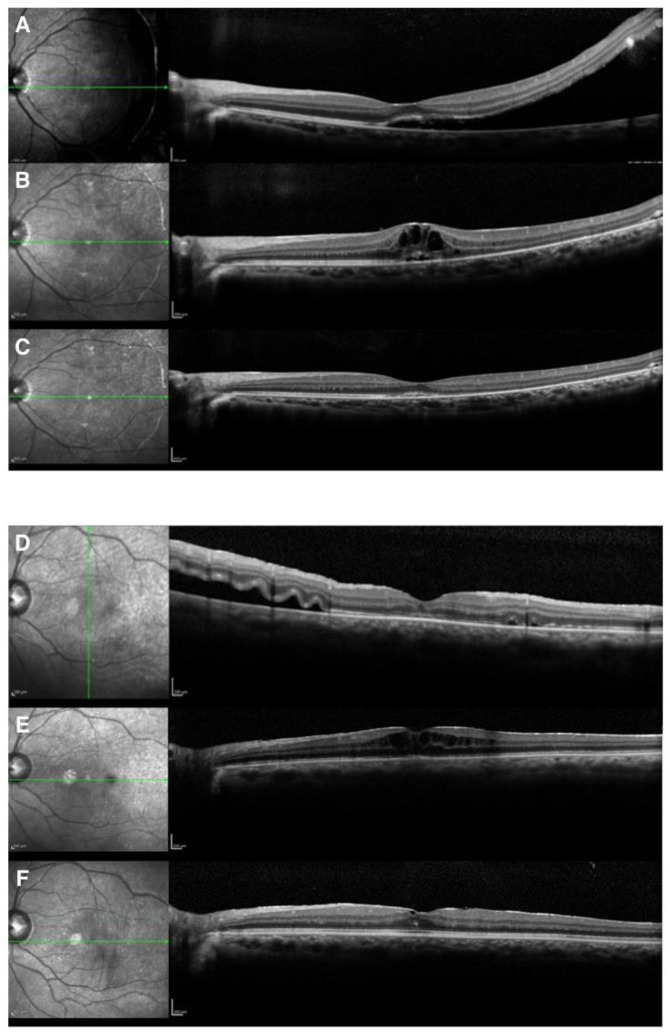
Cases 1 and 2 are presented as representative cases. Case 1: preoperative OCT (**A**); postoperative CME (**B**); OCT at 1 year after surgery (**C**). Case 2: preoperative OCT (**D**); postoperative CME (**E**); OCT at 1 year after surgery (**F**). Both (**A**) and (**D**) showed macular detachment before surgery. As for the timing of onset of postoperative CME, (**B**) showed CME 1 month after surgery, and (**E**) showed CME 2 weeks later. CME disappeared in (**C**) but was found to have remained 1 year after surgery (**F**).

**Table 1 jcm-11-05321-t001:** Baseline clinical characteristics of patients who underwent scleral buckling surgery.

	*n* (%)
Gender	
male	173 (64.8)
female	94 (35.2)
Lens status	
phakic	247 (91.1)
aphakic	1 (0.4)
pseudophakic	23 (8.5)
Macular detachment	
+	109 (40.2)
−	162 (59.8)
No. of breaks	
1	146 (53.9)
2	63 (23.2)
3	32 (11.8)
≧4	30 (11.1)
Extent of retinal breaks	
1 quadrant	232 (85.6)
2 quadrants	33 (12.2)
3 quadrants	5 (1.8)
4 quadrants	1 (0.4)
Extent of retinal detachment	
1 quadrant	106 (39.1)
2 quadrants	124 (45.8)
3 quadrants	34 (12.5)
4 quadrants	7 (2.6)
LASIK	
+	12 (4.4)
−	259 (95.6)

LASIK, laser in situ keratomileusis.

**Table 2 jcm-11-05321-t002:** Factors associated with cystoid macular edema after scleral buckling surgery.

	No CME	Eyes with CME	*p*-Value
	(*n* = 265)	(*n* = 6)	
Lens			0.0078
aphakic or pseudophakic	21	3	
phakic	244	3	
Macula			0.1864
off	105	4	
on	160	2	
SRF drainage			0.5721
+	163	3	
−	102	3	
Air tamponade			0.4071
+	50	2	
−	215	4	
Encircling			0.4584
+	12	0	
−	253	6	
LASIK			0.4584
+	12	0	
−	253	6	

SRF, subretinal fluid. CME, cystoid macular edema.

**Table 3 jcm-11-05321-t003:** Patients with CME after scleral buckling surgery.

Case	Age	Lens	Macular	Drainage	Air	Encircling	LASIK	Preoperative	Postoperative	Onset
No.	Gender		Detachment		Tamponade			BCVA (logMAR)	BCVA (logMAR)	
1	23/F	phakic	+	−	−	−	−	1.22	0	1 M
2	48/M	pseudophakic	+	−	+	−	−	0.52	0.22	2 W
3	61/M	phakic	−	−	−	−	−	−0.08	0.1	1.5 M
4	69/F	pseudophakic	−	+	−	−	−	1	0.22	1 M
5	66/F	pseudophakic	+	+	−	−	−	1.7	0.1	3 M
6	55/M	phakic	+	+	+	−	−	0.8	1.2	3 M

CME, cystoid macular edema. LASIK, laser in situ keratomileusis. BCVA, best-corrected visual acuity. Onset, time of appearance of CME after scleral buckling surgery. M, month(s). W, weeks.

## Data Availability

No applicable.

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
