# Peer review of "Surgical Success Rate of Scleral Buckling Surgery and Postoperative Incidence of Cystoid Macular Edema: 10 Years of Experience at a Single Academic Hospital"

_jcm, 2022, doi:10.3390/jcm11185321_

Round 1
Reviewer 1 Report
In this paper the Authors aimed to to report surgical success rate of scleral buckling surgery (SB) 16 for rhegmatogenous retinal detachment (RRD) in a single academic hospital, analyze the incidence 17 of cystoid macular edema (CME) after SB using optical coherence tomography (OCT), Even though the idea this paper is based on may be of interest, I have major concerns:
1. The paper should be slightly revised to improve its writing.
2. A major issue with this (and other surgical studies) is the inability to isolate variables and the heterogeneity of patients included and amount of surgery performed; the authors should state this as a limitation.
3. For example, one patient may have a single segmental element placed for a single horseshoe tear, and thus theoretically be at lower risk for cystoid macular edema, whereas another may have multiple breaks, require extensive intraoperative ocular manipulation and a high buckle to support the breaks, and this patient would be at higher risk.
4. The results of the study would also be strengthened by having a comparative group, perhaps patients with retinal detachments who underwent pars plana vitrectomy and no encircling element. Otherwise, comparison with published literature (though limited) would be appropriate.
5. What portion of the patient underwent segmental buckles, if so in which quadrants? How many had cryotherapy versus laser and in each which clock hours was the treatment applied to? I would suggest to cite and discuss this paper ( DOI: 10.3390/jcm9041231).
Author Response
RESPONSE TO THE REVIEWERS
We would like to thank the reviewers for their critical comments and useful suggestions. Please find below our point-to-point responses. We have indicated the comments in italicized text and responses in bold. In the revised manuscript, changes are indicated in red.
Reviewer: 1
Comments to the Author
In this paper the Authors aimed to to report surgical success rate of scleral buckling surgery (SB) for rhegmatogenous retinal detachment (RRD) in a single academic hospital, analyze the incidence of cystoid macular edema (CME) after SB using optical coherence tomography (OCT), Even though the idea this paper is based on may be of interest, I have major concerns:
Answer: Thank you for your interest in our paper.
(1). The paper should be slightly revised to improve its writing.
Answer: Thank you for your advice.
(2). A major issue with this (and other surgical studies) is the inability to isolate variables and the heterogeneity of patients included and amount of surgery performed; the authors should state this as a limitation.
Answer: Thank you for your comment. As suggested, we added the following sentence as a limitation in Discussion. “We are aware that different surgical interventions to heterogenous pathologies of RRD may affect occurrence of CME.” (page 8; line 249-250)
(3). For example, one patient may have a single segmental element placed for a single horseshoe tear, and thus theoretically be at lower risk for cystoid macular edema, whereas another may have multiple breaks, require extensive intraoperative ocular manipulation and a high buckle to support the breaks, and this patient would be at higher risk.
Answer: We have revised the Discussion section and added the following text in limitation paragraph (page 8; line 249-250):
“We are aware that different surgical interventions to heterogenous pathologies of RRD may affect occurrence of CME.”
(4). The results of the study would also be strengthened by having a comparative group, perhaps patients with retinal detachments who underwent pars plana vitrectomy and no encircling element. Otherwise, comparison with published literature (though limited) would be appropriate.
Answer: We deeply appreciate the Reviewer’s comment on this point. As suggested, we added comparison with data from published literatures. We revised the Discussion section and added the following text (page 7; line 181–191):
“A recent study comparing surgical outcomes of SB and pars plana vitrectomy for simple phakic macula-on retinal detachment reported that the initial success rate of SB for RRD was 94.9% (130/137), and that of vitrectomy was 84.7% (116/137) [5]. Their study included patients aged <65 years and diagnosed with uncomplicated phakic macula on primary RRD, which is different from our study. However, considering that our initial success rate is 92.7%, which includes patients aged 65 years or older and those with pseudophakic and macula off cases, it seems that surgical efficacy of SB is not inferior to that of vitrectomy. Mete et al. showed that CME after vitrectomy occurred in 6/29 patients (20.7%) in the macula off group and 5/30 (16.7%) in the macula on group, with no difference in CME incidence between the groups (p = 0.69) [6]. We found CME in 6 of 271 patients (2.2%), whose rate seems to be lower than their report.”
(5). What portion of the patient underwent segmental buckles, if so in which quadrants? How many had cryotherapy versus laser and in each which clock hours was the treatment applied to? I would suggest to cite and discuss this paper ( DOI: 10.3390/jcm9041231).
Answer: We wish to express our deep appreciation to Reviewer 1 for these insightful comments. As shown in Table 2, there were 259 cases of segmental buckling and 12 cases of encircling. Segmental buckles were placed in the quadrant(s) where breaks existed. We always use cryopexy but not laser during buckling surgery. Intravitreal gas injections were generally used for the breaks at the superior part, particularly when subretinal fluid remains between breaks and a buckle. We added the suggested reference with discussion. We revised the Discussion section and added the following text in paragraph (page 7; line 228–232):
“A recent study investigating iris vasculature changes following SB in eyes with RRD with anterior-segment optical coherence tomography angiography revealed a uniform reduction of the iris vessel network for the entire 6-month follow-up after SB [15]. Intravitreal changes of cytokines released due to ischemic changes following SB may induce CME. ”
Reviewer 2 Report
The topic of the present study is timely and will be of interest to the readers of the journal. Some images and tables were utilized in order to make the content of the text clearer. Τhe statistical analysis used was appropriate. However, some corrections are suggested to be made.
Corrections of minor importance
-In Methods section, the authors should add specifically which drugs (drug substance or trade name of drug) were administered postoperatively.
Corrections of major importance
- Power analysis should be performed to determine the adequacy of the sample.
Author Response
RESPONSE TO THE REVIEWERS
We would like to thank the reviewers for their critical comments and useful suggestions. Please find below our point-to-point responses. We have indicated the comments in italicized text and responses in bold. In the revised manuscript, changes are indicated in red.
Reviewer: 2
Comments to the Author
The topic of the present study is timely and will be of interest to the readers of the journal. Some images and tables were utilized in order to make the content of the text clearer. Τhe statistical analysis used was appropriate. However, some corrections are suggested to be made.
Answer: Thank you for your interest in our paper.
Corrections of minor importance
-In Methods section, the authors should add specifically which drugs (drug substance or trade name of drug) were administered postoperatively.
Answer: We wish to express our deep appreciation to Reviewer 2 for these insightful comments. As suggested, we have added the details of the medications in the Representative Cases of the Results section as well as the details in other cases. (page 6; line 159–167):
“The treatment courses of CME for the other cases are summarized below. Case 3 presented with CME 1.5 months after SB. After 1 month of oral kallidinogenase and topical prednisolone acetate, CME disappeared. Case 4 had CME 1 month after SB. After 2 weeks of oral acetazolamide and topical NSAIDs, CME did not disappear, and after 4 months of the treatment with topical prednisolone acetate, it disappeared. Case 5 had CME 3 months after SB. After 4 months of the treatment with topical prednisolone acetate, CME disappeared. Case 6 showed CME 3 months after SB. After 1 month of the treatment with topical prednisolone acetate, it disappeared.
Corrections of major importance
- Power analysis should be performed to determine the adequacy of the sample.
Answer: We wish to express our deep appreciation to Reviewer 2 for these insightful comments. We agree with that the sample of eyes undergoing cataract surgery is too small to deduce a conclusion with enough statistical power. If we have to design prospective study with 2 arms which contain either phakic or non-phakic eyes in order to compare the frequency of CME, for example, we need approximately 134 cases in each arm (α=0.05, β=0.02, expected frequency of CME in phakic eyes =0.2, that in non-phakic eyes=0.4). Our current study is a retrospective study and obviously it is not our aim to compare frequency of CME prospectively with lens status. We performed chi-square test as a preliminary analysis to know tendency of CME frequency in eyes with variable lens conditions in a retrospective study. We modified the sentence in a limitation of Discussion as follows. (page 8; line 251–254):
“The number of cases of CME in this study was relatively small in comparison to previous studies, and the occurrence of CME in pseudophakic and aphakic eyes might have been coincidental, which makes statistical power in this study limited.”
Reviewer 3 Report
This is a valuable study from the clinical point of view. However, there are some detailes to be improved. The description of the surgical technique should be extended in methods section- was kryotherapy done, was air given, was subretinal fluid drainage done, was cerclage also performed? It has been described only in the results section.
The treatment of CME should be mentioned in the results section. It is only written in the discussion chapter as follows: “We treated CME with topical NSAIDs, topical prednisolone acetate, subtenon injections of triamcinolone acetonide, intravitreal injections of bevacizumab, or oral acetazolamide and kallidinogenase; however, CME persisted in 1 of 6 eyes.”
Author Response
RESPONSE TO THE REVIEWERS
We would like to thank the reviewers for their critical comments and useful suggestions. Please find below our point-to-point responses. We have indicated the comments in italicized text and responses in bold. In the revised manuscript, changes are indicated in red.
Reviewer: 3
Comments to the Author
This is a valuable study from the clinical point of view. However, there are some detailes to be improved. The description of the surgical technique should be extended in methods section- was kryotherapy done, was air given, was subretinal fluid drainage done, was cerclage also performed? It has been described only in the results section.
Answer: We would like to thank Reviewer 3 for these insightful comments. As suggested, we have revised the Methods section and added the following text (page 2; line 66–71):
“Cryotherapy was used in all the patients to close retinal breaks and diathermy was not used in our institution. Intravitreal gas injection, mostly with air, was generally used in cases of upper retinal breaks particularly when subretinal fluid remains between breaks and a buckle. Subretinal fluid drainage was done in cases with much fluid where buckles could not seal retinal breaks effectively. Encircling was chosen in cases with multiple breaks, in which segmental buckles could not seal all.”
The treatment of CME should be mentioned in the results section. It is only written in the discussion chapter as follows: “We treated CME with topical NSAIDs, topical prednisolone acetate, subtenon injections of triamcinolone acetonide, intravitreal injections of bevacizumab, or oral acetazolamide and kallidinogenase; however, CME persisted in 1 of 6 eyes.”
Answer: We wish to express our deep appreciation to Reviewer 3 for these insightful comments. As suggested, we have revised the Result section and added the following text (page 6; line 159–167):
“The treatment courses of CME for the other cases are summarized below. Case 3 presented with CME 1.5 months after SB. After 1 month of oral kallidinogenase and topical prednisolone acetate, CME disappeared. Case 4 had CME 1 month after SB. After 2 weeks of oral acetazolamide and topical NSAIDs, CME did not disappear, and after 4 months of the treatment with topical prednisolone acetate, it disappeared. Case 5 had CME 3 months after SB. After 4 months of the treatment with topical prednisolone acetate, CME disappeared. Case 6 showed CME 3 months after SB. After 1 month of the treatment with topical prednisolone acetate, it disappeared.”
Round 2
Reviewer 1 Report
i have no further comments.